# Aminoacyl sulfonamide assembly in SB-203208 biosynthesis

Zhijuan Hu[1,2], Takayoshi Awakawa[1,3], Zhongjun Ma[2] & Ikuro Abe [1,3]

Sulfonamide is present in many important drugs, due to its unique chemical and biological properties. In contrast, naturally occurring sulfonamides are rare, and their biosynthetic knowledge are scarce. Here we identify the biosynthetic gene cluster of sulfonamide antibiotics, altemicidin, SB-203207, and SB-203208, from *Streptomyces* sp. NCIMB40513. The heterologous gene expression and biochemical analyses reveal unique aminoacyl transfer reactions, including the tRNA synthetase-like enzyme SbzA-catalyzed L-isoleucine transfer and the GNAT enzyme SbzC-catalyzed β-methylphenylalanine transfer. Furthermore, we elucidate the biogenesis of 2-sulfamoylacetic acid from L-cysteine, by the collaboration of the cupin dioxygenase SbzM and the aldehyde dehydrogenase SbzJ. Remarkably, SbzM catalyzes the two-step oxidation and decarboxylation of L-cysteine, and the subsequent intramolecular amino group rearrangement leads to N-S bond formation. This detailed analysis of the aminoacyl sulfonamide antibiotics biosynthetic machineries paves the way toward investigations of sulfonamide biosynthesis and its engineering.

[1] Graduate School of Pharmaceutical Sciences, The University of Tokyo, 7-3-1 Hongo, Bunkyo-ku, Tokyo 113-0033, Japan. [2] Ocean College, Zhejiang University, Zhoushan 316000, China. [3] Collaborative Research Institute for Innovative Microbiology, The University of Tokyo, Yayoi 1-1-1, Bunkyo-ku, Tokyo 113-8657, Japan. These authors contributed equally: Zhijuan Hu, Takayoshi Awakawa. Correspondence and requests for materials should be addressed to T.A. (email: awakawa@mol.f.u-tokyo.ac.jp) or to I.A. (email: abei@mol.f.u-tokyo.ac.jp)

Actinomycetical alkaloids exhibit remarkable structural diversity, as represented by peptide, aminobenzoate, oxazole, thiazole, and indolocarbazole-derived secondary metabolites[1–5]. Their biosynthetic pathways are rich in rare enzymes, including those that catalyze heteroatom–heteroatom (X–X) bond-forming reactions, thereby appending important biological functions to the molecules[6]. While several enzymes involved in the N–N bond formation have been identified recently; e.g., CreDEM in cremeomycin biosynthesis[7,8], KtzIT in piperazate biosynthesis[9], and Spb38 and 40 in s56-p1 biosynthesis[10], only two examples have been reported for N–S bond formation. Thus, the FAD-dependent monooxygenase XiaH catalyzes the radical-mediated N–S bond-forming reaction in the biosynthesis of sulfadixiamycins A[11], and the putative aminotransferase AcmN was proposed to be involved in the N–S bond formation in 2-deschloro-dealanylascamycin biosynthesis[12]. Considering the presence of hundreds of N–S bond containing alkaloids in nature[5], many unusual enzyme reactions remain unexplored for the N–S bond-forming chemistries.

The sulfonamide functional group exhibits less basicity and more rigid structural properties, as compared with carboxamide, and thereby contributes to biological activities such as sweeteners, and drugs functioning as diuretics, uricosurics, hypoglycemic treatments, antimicrobial agents, and herbicides[13]. This functional group is widely used in medicinal chemistry through the coupling methodology of sulfonyl chloride and amide, whereas its biosynthesis still remains to be rigorously investigated. For example, altemicidin (1), consisting of sulfonamide and 6-azatetrahydroindane moieties, was first isolated from *Streptomyces sioyaensis* SA-1758 as an insecticidal and acaricidal antibiotic[14,15]. In contrast, its aminoacylated derivatives, SB-203207 (2) and SB-203208 (3), were isolated from *Streptomyces* sp. NCIMB 40513 as isoleucyl tRNA synthetase inhibitors[16,17] (Fig. 1a). Their unique chemical structures and biological activities have attracted the interest of synthetic chemists, leading to the recent total syntheses of 1 and 2[18–20]. In contrast, since no biosynthetic studies have been reported, we conducted investigations of the biogenesis of 1–3 in *Streptomyces* sp. NCIMB 40513.

Here, we report the identification of the biosynthetic gene cluster of 1–3, and its successful heterologous expression in the *Streptomyces lividans* host. Furthermore, biochemical analyses disclose three aminoacyl transfer reactions and the biogenesis of sulfonamide through N–S bond formation. Thus, we elucidate the three aminoacyl transfer reactions, including the tRNA synthetase-like enzyme SbzA-catalyzed installation of L-isoleucine onto 1 to afford 2, and the biogenesis of a sulfonamide from L-cysteine by the cupin dioxygenase SbzM. These unusual biosynthetic machineries lead to the construction of the rare aminoacyl sulfonamide molecular scaffold.

## Results

**Identification of the biosynthetic gene cluster**. First, we sought to identify the biosynthetic gene cluster of SB-203208 (3), an Ile-tRNA synthetase inhibitor, with one sulfonamide, two carboxamides, one ester, L-isoleucine, and β-methylphenylalanine partial structures (Fig. 1a), from the genome sequence of *Streptomyces* sp. NCIMB 40513. Since the resistance gene for a specific antibiotic is often clustered with its biosynthetic genes, we focused on the Ile-tRNA synthetase genes in the genome. As a result, a BLASTp search using *Escherichia coli* Ile-tRNA synthetase (accession number P00956.5) as the query revealed two Ile-tRNA synthetase genes, *Ssp-IleRS* (27% amino acid identity) and *sbzA* (25% identity). Interestingly, *sbzA* is located in a 20-kb gene cluster (*sbz* cluster) encoding 18 genes, including those of two adenylate-forming enzymes (*sbzB* and *sbzL*) and two acyl carrier proteins (*sbzG* and *sbzK*), which are thought to be involved in

amide bond-forming reactions[21] (Fig. 1b and Supplementary Table 1). In addition, the *sbz* cluster includes one gene (*sbzD*) that exhibits 40% identity to MppJ, a methyltransferase that produces β-methylphenylalanine in mannopeptimycin biosynthesis[22]. Based on these observations, we assigned the *sbz* cluster as a candidate biosynthetic gene cluster of 1–3. Interestingly, the *sbz* cluster consists of two operons, *sbz2* (*sbzA-C*) and *sbz1* (*sbzD-R*), and thus it is suitable for heterologous expression since only two promoters are sufficient for the expression of all of the genes.

**Heterologous expression of the *sbz* cluster**. We cloned the *sbz1* and *sbz2* operons under the control of the *ermE* promoter on two phage-integration vectors, and introduced them into the *Streptomyces lividans* TK21 strain (Supplementary Data set 1). First, we analyzed the water extract of the *S. lividans* culture expressing *sbz1* by HPLC, and found one peak at $R\mathrm{t} = 8.0$ min that was not present in the negative control (Fig. 2). The product 1 (10 mg/L) was thus purified from the large-scale culture. By comparing its NMR and HR-MS data with those in the literature[18], we identified the product as altemicidin (1) (Supplementary Figs. 1–3). Next, we examined the extract from *S. lividans* expressing both the *sbz1* and *sbz2* operons, and detected 2 (1.1 mg/L) and 3 (1.8 mg/L) as their products (Fig. 2). Comparisons of the NMR data unambiguously established that they are identical to SB-203207 (2) and SB-203208 (3), respectively[17] (Supplementary Figs. 4–8). The compound 4 whose *m/z* was consistent with β-methylphenyalanyl-1 was also detected as a product (Supplementary Fig. 1). In addition, we analyzed the β-methylphenylalanine hydrolyzed from 3 by chiral HPLC, and confirmed its stereochemistry as 2S, 3R[23] (Supplementary Fig. 9). These data clearly demonstrated that 1 was produced by the enzymes encoded in the *sbz1* operon, and the enzymes encoded in the *sbz2* operon were required for the installation of L-isoleucine and β-methylphenylalanine onto 1, to yield 2 and 3.

**Enzymes involved in aminoacyl transfer reactions**. To understand the functions of the three enzymes encoded in the *sbz2* operon, the purified His-tagged recombinant proteins of SbzA (Ile-tRNA synthetase-like enzyme), SbzB (adenylate-forming enzyme), and SbzC (GNAT family enzyme) (Supplementary Fig. 10, Supplementary Table 1) were subjected to in vitro enzyme reactions. The results demonstrated that SbzA efficiently catalyzes the *N*-aminoacyl transfer reaction of L-isoleucine onto the primary sulfonamide of 1 to yield 2, in the presence of Ile-tRNA$_{E.coli}^{Ile}$, which was supplied by an in vitro translation kit based on an *E. coli* lysate (Fig. 3a, d). The addition of RNase to the reaction mixture completely abolished the production of 2 (Fig. 3a), and thus it suggested that Ile-tRNA is essential for the SbzA enzyme reaction. We also conducted the SbzA enzyme reaction with 1 and Ile-tRNA$_{Ssp}^{Ile}$, which was synthesized by Ssp-IleRS from bulk tRNA isolated from *Streptomyces* sp. NCIMB40513, and detected 2 as a product (Supplementary Fig. 11). This result clearly reconfirmed the dependency of SbzA reaction on Ile-tRNA. Interestingly, SbzA alone can also produce a trace amount of 2 in the absence of Ssp-IleRS, even though the yield is 94% less than SbzA + Ssp_IleRS reaction. These data suggested that SbzA can also catalyze the isoleucyl transfer reaction onto tRNA$^{Ile}$, but with much less efficiency than that of Ssp_IleRS.

We then investigated the enzymes involved in the *O*-acyltransfer of β-methylphenylalanine onto 2 to yield the final product 3 (Fig. 3d). We thus tested the adenylate-forming enzyme SbzB with various amino acids, including β-methylphenylalanine, as substrates[24]. As a result, SbzB indeed selectively adenylated (2S, 3R)-β-methylphenylalanine as the best substrate

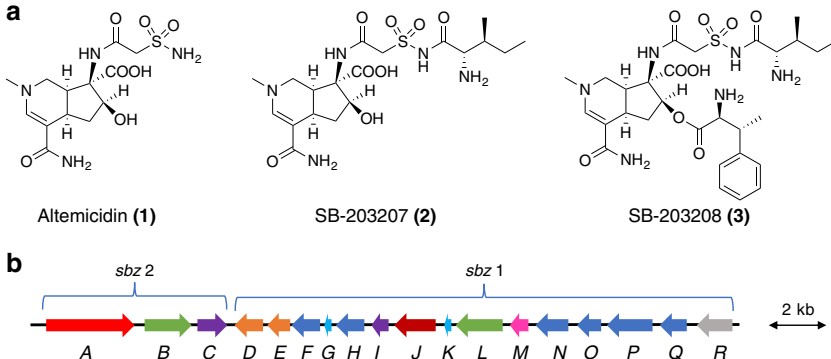

**Fig. 1** Chemical structure of the aminoacyl sulfonamide antibiotics and the gene organization of *sbz* cluster. **a** Structures of the aminoacyl sulfonamide antibiotics and **b** the organization of the *sbz* cluster (Please see Supplementary Table 1 for more details.)

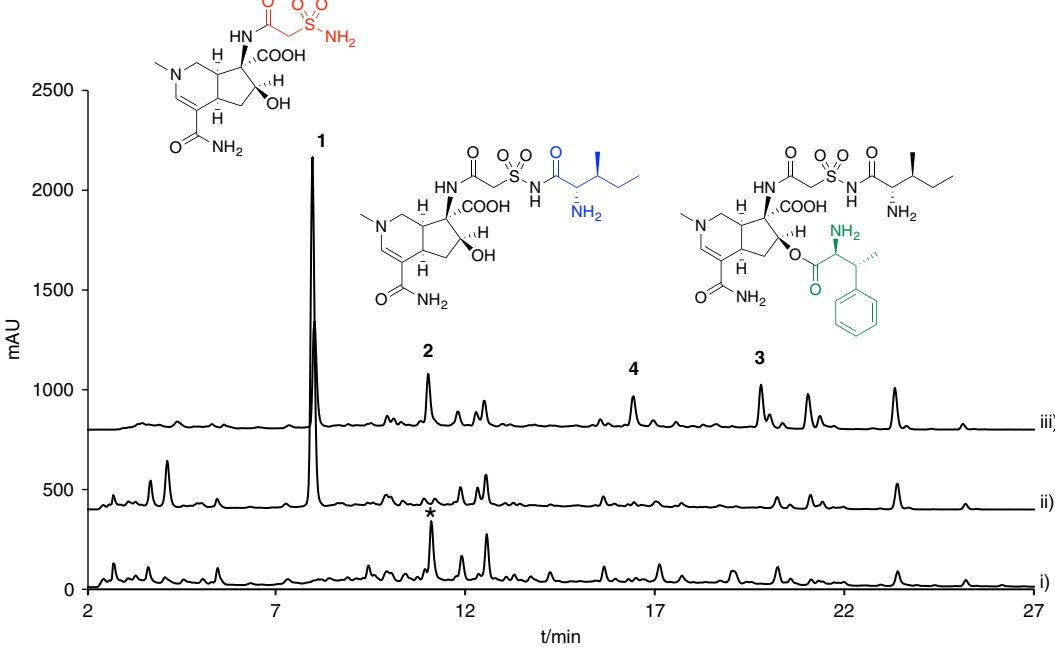

**Fig. 2** HPLC analysis (monitored at 300 nm) of the water extracts from the *S. lividans* strain expressing *sbz* gene cluster. The HPLC traces of the extracts from i) *S. lividans* TK21, ii) *S. lividans* TK21/*sbz1*, and iii) *S. lividans* TK21/*sbz1&2*. * indicates the compound with 275 nm UV max which is not relevant to **1–4**

(Supplementary Fig. 12), whereas other stereoisomers of β-methylphenylalanine were not accepted for the enzyme reaction (Supplementary Fig. 9). Since the candidate acyltransferase SbzC (GNAT family enzyme) shares similarity with mycothiol acetyltransferase, which accepts acetyl-CoA[25], we incubated SbzB and SbzC with **2**, ATP, and CoA, anticipating that the CoA thioester of β-methylphenylalanine thus formed may serve as a substrate for the SbzC-catalyzed acyltransfer reaction. However, the reaction did not yield the expected product **3** (Fig. 3b), suggesting that the adenylate-forming enzyme SbzB utilizes an acyl carrier protein to load β-methylphenylalanine, as in the case of the NRPS system. Indeed, when we added an acyl carrier protein, SbzG or SbzK, encoded in the *sbz1* operon (Supplementary Table 1), to the assay mixture, the enzyme reaction afforded the β-methylphenylalanine-loaded SbzG or SbzK carrier protein, as confirmed by an LC-ESI-MS analysis (Supplementary Fig. 13). Interestingly, SbzB accepted both SbzG and SbzK with almost the same preference in vitro, and the subsequent *O*-acyltransfer reaction by SbzC successfully yielded the small amount of final product **3** (Fig. 3b, d). We also tested **1** as an acceptor substrate

for SbzBCG reaction, to detect **4** as a product, but with much less efficiency (Supplementary Fig. 14).

Finally, the second adenylate-forming enzyme SbzL, encoded in the *sbz1* operon (Supplementary Table 1), was shown to be responsible for the *N*-acyltransfer of 2-sulfamoylacetic acid (**5**) onto **6** to yield altemicidin (**1**) (Fig. 3d). SbzL thus selectively accepts **5** as a substrate, while the other tested amino acids, including L-cysteine, L-cysteic acid, and L-cysteine sulfinic acid, were not the substrates of SbzL (Supplementary Fig. 12). As in the case of SbzB, SbzL also accepted both SbzG and SbzK as acyl carrier proteins (Supplementary Fig. 13). Furthermore, in vitro analyses clearly established that the second GNAT *N*-acyltransferase, SbzI, installs 2-sulfamoylacetate from the carrier protein onto the amino group of **6** to yield **1** (Fig. 3c, d). This suggested that the sulfonamide **5** is biosynthesized before it is loaded onto the acyl carrier protein. Notably, most of the GNAT family acyltransferases accept CoA-linked substrates[26], but some known exceptions include the *N*-acyl amino acid synthase FeeM, which uptakes its substrates from the acylated carrier protein prepared by adenylate-forming enzymes[27]. This

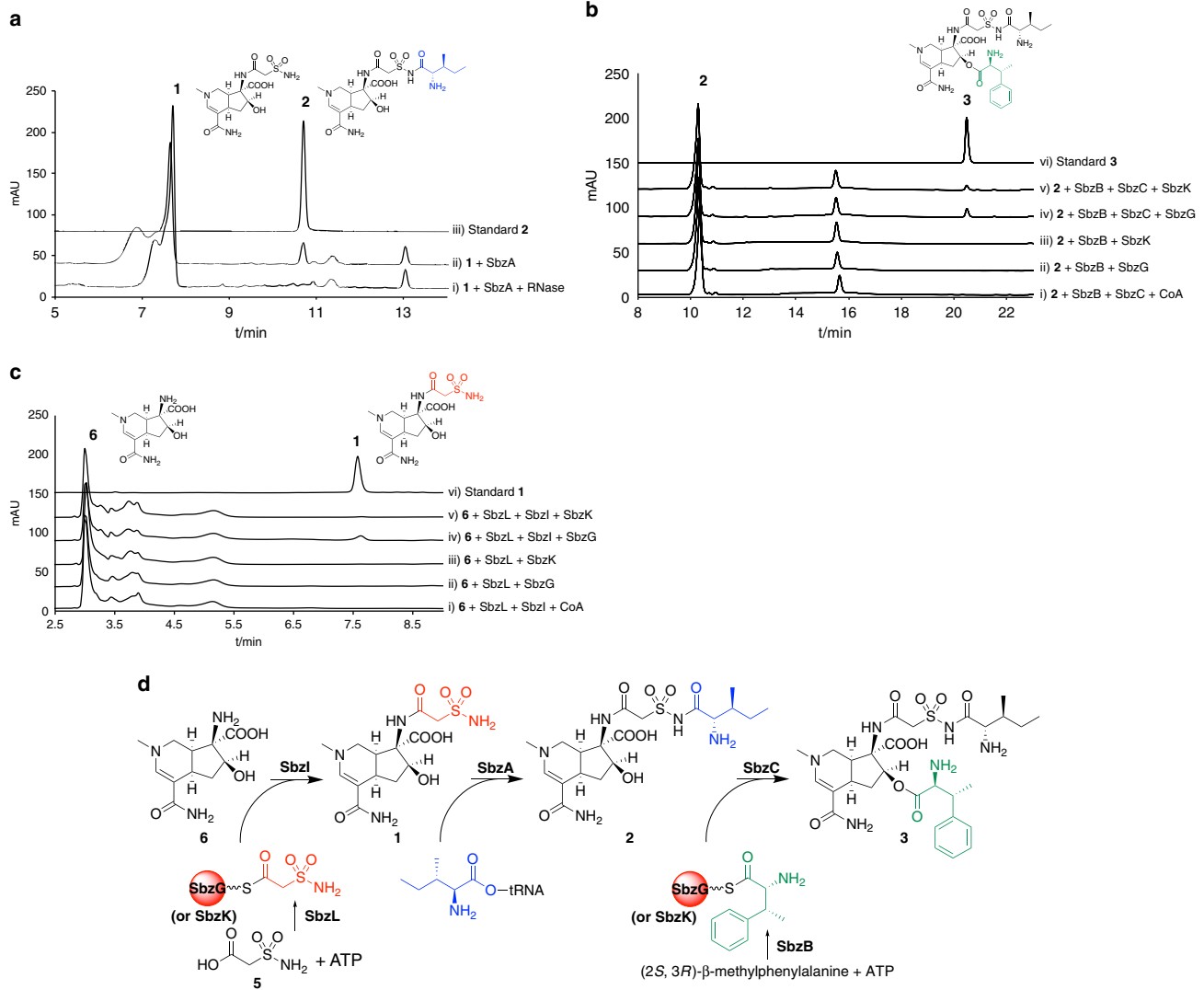

**Fig. 3** HPLC analysis (monitored at 300 nm) of the in vitro aminoacyl transfer enzyme reactions. **a** The assay of SbzA with altemicidin (**1**) and Ile-tRNA as substrates; (i) SbzA + RNase, (ii) SbzA, and (iii) the authentic standard of SB-203207 (**2**). **b** The assay of SbzBCG (or BCK) with **2** and β-methylphenylalanine as substrates; (i) SbzB + SbzC + CoA, (ii) SbzB + SbzG, (iii) SbzB + SbzK, (iv) SbzB + SbzC + SbzG, (v) SbzB + SbzC + SbzK, and (vi) the authentic standard of SB-203208 (**3**). **c** The assay of SbzILG (or ILK) with **6**; (i) SbzL + SbzI + CoA, (ii) SbzL + SbzG, (iii) SbzL + SbzK, (iv) SbzL + SbzI + SbzG, (v) SbzL + SbzI + SbzK, and (vi) the authentic standard of **1**. **d** Proposed mechanisms of the aminoacyl transfer enzyme reactions by SbzABCGIL (or SbzABCILK)

would probably increase the specificity of substrate recognition, as in the cases of SbzC and SbzI.

**Enzymes involved in biogenesis of sulfonamide**. To elucidate the biosynthesis of sulfonamide and the N–S bond-forming reaction, we tested the gene deletions of *sbzF, H, J*, and *M–Q* (Supplementary Table 1). As a result, all of the mutant strains no longer produced **1** (Supplementary Fig. 15a), indicating that all of the tested genes are essential for the production of **1**. The mutants were further analyzed for their metabolites by Hilic–LC–MS. While the Δ*sbzF, H*, and *N–Q* mutants still yielded **5**, Δ*sbzM* and Δ*sbzJ* no longer produced it (Supplementary Fig. 15b). In addition, when **5** was supplemented into the Δ*sbzJ* and Δ*sbzM* culture media, both of the mutants resumed the production of **1** (Supplementary Fig. 15c). These results confirmed that SbzM (cupin dioxygenase) and SbzJ (aldehyde dehydrogenase) (Supplementary Table 1) are essential for the biogenesis of the sulfonamide **5**.

We then performed feeding experiments by supplying [$^{13}C_3$, $^{15}N_1$]-L-cysteine to the culture medium, and analyzed the NMR

spectra of **1**. As a result, the $^{13}C$ NMR signals at C-12 and C-13 and the $^{15}N$ NMR signal (δ 89.5), corresponding to the primary amine, were significantly enhanced (Supplementary Fig. 16a–c), which clearly supported the incorporation of cysteine into sulfonamide. Interestingly, the m/z values of **1** and **5** from the fed culture were 3 Da larger than the original one (Supplementary Fig. 16d, e). This indicated that the $^{15}N$ atom of the labeled cysteine was retained in the sulfonamide, as were the two $^{13}C$ atoms after the decarboxylation reaction. Based on these observations, we propose that L-cysteine undergoes the sequential oxidation of the sulfur atom and the decarboxylation. The subsequent intramolecular rearrangement of the amino group onto the oxidized sulfur atom leads to the S–N bond formation, during the enzyme reactions catalyzed by SbzM (cupin dioxygenase) and SbzJ (aldehyde dehydrogenase) (Fig. 4d).

To further clarify the biosynthesis of sulfonamide, we performed in vitro enzyme reactions of SbzJ and SbzM with L-cysteine as the substrate. To monitor the enzyme reaction, the carboxyl groups were derivatized with 4-bromophenacyl (4-BPA)

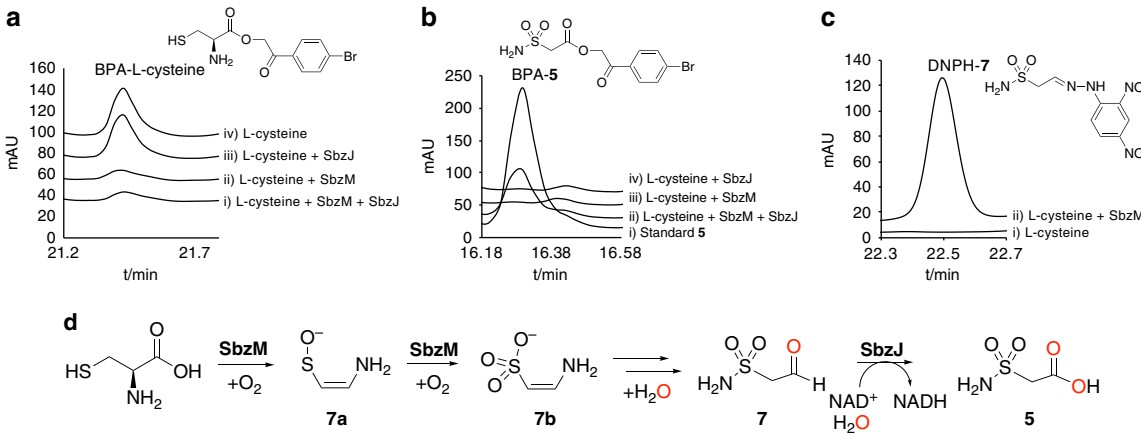

**Fig. 4** HPLC analysis of the products of SbzM and SbzJ reactions. The substrate L-cysteine was derivatized with 4-bromophenacyl (4-BPA) bromide. **a** (i) SbzM+SbzJ, (ii) SbzM, (iii) SbzJ, and (iv) no enzyme, and the reaction products were derivatized with 4-BPA. **b** (i) Standard **5**, (ii) SbzM + SbzJ, (iii) SbzM, and (iv) SbzJ, or 2,4-dinitrophenylhydrazine (DNPH) hydrochloride. **c** (i) No enzyme and (ii) SbzM. The chromatogram represents the UV absorbance at 254 nm. **d** It was demonstrated that L-cysteine is converted into 2-sulfamoylacetic acid (**5**) by the collaboration of SbzM and SbzJ. In contrast, the cupin dioxygenase SbzM produces 2-sulfamoylacetic aldehyde (**7**) from L-cysteine via (Z)-(2-aminovinyl)sulfanolate (**7a**) and (Z)-2-aminoethanone-1-sulfonate (**7b**). The oxygens derived from water are colored red

bromide, and the results clearly demonstrated that L-cysteine is efficiently converted into the sulfonamide **5** by the collaboration of SbzM and SbzJ (Fig. 4a, b). Next, we analyzed the cupin dioxygenase SbzM enzyme reaction product, 2-sulfamoylacetic aldehyde (**7**), by 2,4-dinitrophenylhydrazine (DNPH) hydrochloride derivatization (Fig. 4c) (Supplementary Figs. 1 and 17–21, Supplementary Table 2). To unambiguously determine the product, the SbzM enzyme reaction was performed in an NMR tube with $^{13}$C-enriched $[^{13}C_3, ^{15}N_1]$-L-cysteine as the substrate, and analyzed by $^{13}$C NMR, DEPT135, and HMQC spectroscopy. The analyses detected a single product that possessed one methylene ($\delta^1$H 3.40, $\delta^{13}$C 60.2 (d, $J = 179.5$ Hz)) and one methine ($\delta^1$H 5.34, $\delta^{13}$C 86.4 (d, $J = 179.5$ Hz)), and its $m/z$ was 140.0022 (Cal. 140.0023, $C_2H_6NO_4S^-$) (Supplementary Figs. 22–25). These data indicated that the detected product was a geminal diol[28], which is the hydrate form of the sulfonamide aldehyde **7**.

Taken together, for the biosynthesis of the sulfonamide **5**, we propose that the cupin dioxygenase SbzM catalyzes the decarboxylation coupled with monooxygenation to yield (Z)-(2-aminovinyl)sulfanolate **7a** and dioxygenation of L-cysteine to yield the reactive sulfonate–enamine **7b**. The subsequent N–S bond-forming intramolecular rearrangement of the amino group, and the oxidation of the resulting sulfonamide aldehyde **7** by the aldehyde dehydrogenase SbzJ, finally generate **5** (Fig. 4d). Based on the annotation of the aldehyde dehydrogenase SbzJ and the fact that we could not detect any additional intermediates from the SbzJ/SbzM enzyme reaction in the absence of NAD$^+$ (Supplementary Fig. 26), we concluded that SbzJ catalyzes only the dehydrogenation of the aldehyde **7** to yield **5**. The structure of **7b** was determined as (Z)-2-aminoethanone-1-sulfonate, from the structure of the dansylated-**7b** (Supplementary Figs. 1, 27–30, Supplementary Table 3).

When L-cysteine was incubated with SbzM and SbzJ in the presence of H$_2$$^{18}$O, the $m/z$ of **5** increased by 4 Da (Supplementary Fig. 31), indicating that the second H$_2$$^{18}$O molecule was incorporated in addition to the one incorporated in the SbzJ aldehyde dehydrogenase reaction. Furthermore, the LC–MS analysis of the SbzM enzyme reaction product with $m/z =$ 121.9912, corresponding to the aldehyde **7**, was $^{18}$O-labeled when we employed H$_2$$^{18}$O as the solvent (Supplementary Fig. 32). These data clearly suggested that **7** was generated through the hydrolysis of **7b** (Figs. 4d, 5c).

Finally, since the presence of a metal ion is reported to be crucial for the cupin enzyme activity, we examined the metal ion dependency of the SbzM enzyme reaction by the metal chelation and restoration method[28]. Thus, after an incubation with EDTA, various metal ions (Fe$^{2+}$, Cu$^{2+}$, Zn$^{2+}$, Mn$^{2+}$, Ni$^{2+}$, and Co$^{2+}$) were added and the enzyme activities were evaluated. As a result, Ni$^{2+}$, Cu$^{2+}$, and Fe$^{2+}$ restored the activity to 52, 45, and 40%, respectively (Supplementary Fig. 33). Furthermore, we confirmed that the contents of Ni$^{2+}$ and Fe$^{2+}$ increased while no increase for other metal ions in each of the enzymes by ICP-MS analyses. Therefore, we concluded that Fe$^{2+}$ and Ni$^{2+}$ bind to the enzyme and are essential for the SbzM enzyme reaction. On the other hand, the enzyme incubated with Cu$^{2+}$ precipitated after the successive dialysis to reduce the concentration of metal to significantly lower than the enzyme concentration. Notably, the preference for the Ni ion is rare for the cupin enzymes, but was also reported for the dimethylsulfoniopropionate lyase DddK[29,30] and Streptomyces quercetinase QueD[31,32]. The $K_M$ and k$_{cat}$/$K_M$ values for L-cysteine were $0.53 \pm 0.08$ mM and $1.67 \times 10^3$ min$^{-1}$mM$^{-1}$ ($n = 2$) (Supplementary Fig. 33), respectively, which are comparable with those of other cupin family enzymes[29].

## Discussion

In this study, we identified the biosynthetic gene cluster of the antibiotics with rare sulfonamide and 6-azatetrahydroindane scaffolds, altemicidin (**1**), SB-203207 (**2**), and SB-203208 (**3**), from Streptomyces sp. NCIMB40513, and succeeded in heterologous gene expression in S. lividans TK21. Furthermore, in vitro biochemical analyses revealed three unique aminoacyl transfer enzyme reactions, including that of the GNAT family enzyme SbzI-catalyzed installation of 2-sulfamoylacetic acid (**5**) onto the 6-azatetrahydroindane precursor to yield **1**, the tRNA synthetase-like enzyme SbzA-catalyzed L-isoleucine transfer to yield **2**, and the second GNAT enzyme SbzC-catalyzed β-methylphenylalanine transfer to yield the final product **3** (Fig. 3d). Moreover, we elucidated the N–S bond-forming reaction in the biogenesis of the sulfonamide **5** from L-cysteine, by collaboration of the cupin dioxygenase SbzM and the aldehyde dehydrogenase SbzJ (Fig. 4d). These unusual biosynthetic machineries successfully led to the construction of the rare aminoacyl sulfonamide molecular scaffold.

Remarkably, SbzA is a tRNA synthetase-like enzyme that catalyzes aminoacyl transfer from the corresponding aminoacyl-tRNA (aa-tRNA) to a non-peptide secondary metabolite. A homology model of SbzA constructed with Swiss-Prot, using Ile-tRNA synthetase from *Thermus thermophilus*[33,34] (1JZQ, 37.6% identity) as the template, showed that it belongs to the class Ia aa-tRNA synthetases, which consist of multiple domains, including the Rossmann fold, editing, and zinc-binding domains (Supplementary Fig. 34)[33–35]. The in vitro assay of SbzA with bulk tRNA showed that Ssp_IleRS aminoacylates tRNA[Ile], and then, SbzA installs isoleucyl moiety on **1**. Thus, we concluded that Ssp_IleRS catalyzes the tRNA aminoacylation, and SbzA in the gene cluster catalyzes the aminoacyl transfer from Ile-tRNA onto **1** to yield **2**. Interestingly, SbzA can also catalyze aminoacylation of tRNA[Ile], but with much less efficiency than that of Ssp_IleRS. Notably, SbzA possesses conserved "HIGH" and "KMSKS" sequences that are signature ATP-binding motifs in the Rossmann fold of class I aaRS[35], even though there are slight modifications (Supplementary Fig. 34), which could support the observation that SbzA also catalyzes the tRNA acylation. The detailed structure–function relationship on SbzA should be the subject of our future study. The aminoacyl moieties introduced by the known aa-tRNA-dependent transferases in secondary metabolism are Ser, Leu, Gly, Ala, Phe, Tyr, and Trp, and this is a report of the installation of Ile by a tRNA-dependent aminoacyl transferase[36].

In the biogenesis of a sulfonamide, we proposed that the cupin dioxygenase SbzM catalyzes the sequential two-step oxidations of L-cysteine to yield the reactive (*Z*)-2-aminoethanone-1-sulfonate (**7b**) via (*Z*)-(2-aminovinyl)sulfanolate (**7a**), and the subsequent intramolecular rearrangement of the amino group onto the sulfur atom (N–S bond formation) generates the sulfonamide aldehyde **7** (Fig. 4d).

The reaction from L-cysteine to **7a** was hypothesized as below (Fig. 5a). The cysteine binding to the enzyme changes the coordination of the iron, followed by the binding of the molecular oxygen to the ferrous ion and successive nucleophilic addition to yield **7a–1**, in a similar manner to cysteine dioxygenase[37]. The resultant oxide anion abstracts the Cβ proton and the carbon–sulfur double bond is newly formed in **7a–2**. This Cβ proton might be supported through ionic interaction with the amino acid residues in SbzM as seen in EpiD, a flavin-dependent cysteine decarboxylase[38]. An evolving negative charge on sulfur

moves to oxygen, resulting in **7a–3** sulfene structure, which is generated through elimination of HCl from sulfinyl chloride in chemical synthesis[39]. Finally, this intermediate generates a more stable ene-thioaldehyde S-oxide[40] **7a**.

The oxidation mechanism of **7a** to **7b** is predicted to resemble that proposed based on the the X-ray structure of the cupin-type cysteine dioxygenase complexed with cysteine persulfenate[37] (Fig. 5b). The ferrous-bound molecular oxygen is attacked by the electron from the sulfanolate in nucleophilic addition mechanism to give **7b–1**. The addition of the negatively charged distal oxygen to the sulfoxide generates thiadioxirane **7b–2**. The migration of the lone pair on the sulfur driven by the electron migration from the sulfanolate and the cleavage of the O–O bond yield **7b**. However, we cannot exclude the possibility of the S–O bond formation through radical mechanism in this step, as discussed in the literature[37]. Finally, **7b** tautomerizes to **7c**, the resultant imine of **7c** is hydrolyzed, and the hydrolyzed amine in **7d** attacks the protonated sulfonic acid, resulting in **7** accompanied with elimination of water as a leaving group (Fig. 5c).

The multistep reaction from L-cysteine to **7** triggered by SbzM is intriguing, as such a cascade reaction initiated by a cupin oxidase has rarely been reported, except for the one catalyzed by the recently reported fungal cupin enzyme in phenalenone biosynthesis[41]. The cupin enzyme that oxidizes the sulfur of cysteine[42] and the one that abstracts the methylene proton at the α-position of dimethylsulfoniopropionate[29,30] were known; however, SbzM is a cupin-type cysteine decarboxylase. Oxalate decarboxylase is a cupin-type decarboxylase[43], but its reaction mechanism is likely to be different from the one that we proposed. The structural studies on SbzM are now in progress to further clarify its reaction mechanism.

It should be noted that the BLASTp search in JGI (https://img.jgi.doe.gov/) and NCBI (https://blast.ncbi.nlm.nih.gov/Blast.cgi) indicated that there are four paralog gene clusters in *Streptomyces melanosporofaciens* DSM 40318, *Streptomyces* sp. NRRL S-1868, *Streptomyces antioxidans* MUSC 164, and *Streptomyces* sp. NRRL F-5053 (Supplementary Fig. 35). In addition, homologs of the cupin dioxygenase SbzM are encoded in the genomes of several bacteria, and interestingly, some of them are clustered with secondary metabolism-like genes (Supplementary Figs. 35 and 36), suggesting the distribution of the SbzM-catalyzed cysteine metabolism that generates the reactive sulfonate–enamine compounds.

**Fig. 5** The proposed reaction mechanism of SbzM. **a** L-cysteine is transformed into (*Z*)-(2-aminovinyl)sulfanolate (**7a**) via the decarboxylation coupled with sulfur monooxygenation. **b 7a** is dioxygenated into (*Z*)-2-aminoethanone-1-sulfonate (**7b**). **7b** was tautomerized into **7c**, the hydration of the imine of **7c**, and the hydrolyzed amine in **7d** attacks the protonated sulfonate to yield **7** with the release of water. The Fe can be substituted with Ni. The oxygens derived from water are colored red

In conclusion, this study identified the biosynthetic gene cluster for the class of alkaloids with a 6-azatetrahydroindane and sulfonamide structure, and illuminated the biosynthetic machineries for the biogenesis of sulfonamide antibiotics. Importantly, the cupin dioxygenase SbzM catalyzes the cysteine-processing reactions to generate a sulfonamide, which is further modified by successive and unusual aminoacyl transfers. This knowledge will pave the way toward investigations of sulfonamide biosynthesis, as well as its engineering, to produce biologically active, unnatural sulfonamide antibiotics.

## Methods

**General experimental procedures.** Solvents and chemicals were purchased from Sigma-Aldrich, Wako Chemicals Ltd., or Kanto Chemical Co., Inc., unless noted otherwise. Oligonucleotide primers were purchased from Eurofins Genetics or Sigma-Aldrich. PCR was performed using a TaKaRa PCR Thermal Cycler Dice® Gradient (TaKaRa), with Prime STAR Max DNA Polymerase (TaKaRa). Sequence analyses were performed by Eurofins Genetics. HPLC analysis was performed on a Shimadzu Prominence HPLC system with a Separar C18G column (4.6 mm I.D. × 250 mm, Rikaken Co. Ltd., Nagoya, Japan). The LC–MS analysis was performed on a Bruker Compact qTOF mass spectrometer with a Shimadzu Prominence HPLC system, using a COSMOSIL 5C18-AR-II column (2.0 mm I.D. × 150 mm, Nacalai Tesque, Inc., Kyoto, Japan). NMR spectra were obtained with JEOL ECX-500 or ECZ-500 spectrometers. *Streptomyces* sp. NCIMB 40513 was purchased from NCIMB, with the permit from GSK, Inc.

**Genome sequencing.** The sequencing was performed with an Ion PGM sequencer (Life Technologies), with a total number of 2,953,291 sequence reads (~300 bp). These sequences were assembled using the de novo assembler MIRA (v3.4.2.0). Further assembly to produce larger contigs was achieved with the Geneious assembler (Biomatters), with the default medium sensitivity. The putative protein-coding sequences (CDSs) were determined by a combination of 2ndFind (http://biosyn.nih.go.jp/2ndfind/).

**Heterologous expression in *Streptomyces lividans* TK21.** The proposed gene cluster was divided into two parts, *sbzD-R* as sbz 1 and *sbzA-C* as sbz 2. The sbz 1 part was amplified from the genome of *Streptomyces* sp. NCIMB 40513 with the primers I-VI (Supplementary Data set 1) into three fragments, and inserted into the *Eco*RI and *Hin*dIII sites of the pUC19 vector by in-fusion cloning. The fragment comprising φBT1, the *aac(3)IV* apramycin resistance gene, and the *ermE* promotor was amplified with the primers VII-VIII and inserted into the pUC19-sbz 1 vector by λ RED-mediated recombination with the pKD78 vector. The resultant vector was renamed pZH1-sbz1. The sbz 2 part was amplified with the primers IX-X and inserted into the pUC19 vector, along with φC31, the *tsr* thiostrepton resistance gene, and the *ermE* promotor, by in-fusion cloning. The resultant vector was renamed pZH2-sbz2.

Transformation of the vectors was performed by the general protoplast-PEG mediated transformation for *S. lividans* TK21. The resultant strains were named *S. lividans*/sbz 1 and *S. lividans*/sbz 1 and 2.

**Analysis of metabolites.** Fermentation of the heterologous expression strains was performed in 500 mL-baffled flasks containing 100 mL A3M medium, consisting of 2.0% starch, 2.0% glycerol, 0.5% glucose, 1.5% Pharma media (Archer Daniels Midland Co.), 1.0% HP-20 (Nihon Rensui), and 0.3% yeast extract (pH 7.0), for 6 days at 30 °C and 160 min$^{-1}$. The culture broth was centrifuged and analyzed directly. The HPLC solvent system was $H_2O$ containing 50 mM $NH_4OAc$ (solvent A) and $CH_3OH$ (solvent B), with a gradient of 5–100% B over 30 min at a flow rate of 1.0 mL/min. The target compounds were monitored at 300 nm.

**Isolation and structure elucidation.** Altemicidin **1** (10 mg) was isolated from a 1.0 L culture of the *S. lividans*/sbz 1 strain. The culture broth was freeze-dried and subjected to ODS open-column (Cosmosil 75C18-OPN, Nacalai Tesque, Inc.) chromatography, eluted with 50 mM ammonium acetate. The fractions containing **1** were combined, freeze-dried, and further purified by semi-preparative HPLC with a Triant C18 column (10.0 mm I.D. × 250 mm, YMC, Kyoto, Japan), using 5% $CH_3CN/H_2O$ as the solvent.

Altemicidin (**1**): white powder; UV ($CH_3OH$) $\lambda_{max}$: 300 nm; [a]30$_D$ = +7.5 (c = 0.1, $H_2O$); $^1$H NMR (500 MHz, $D_2O$) δ 7.38 (1H, s), 4.36 (1H, d, $J$ = 14.4 Hz), 4.28 (1H, d, $J$ = 14.4 Hz), 4.28 (1H, overlapping), 2.96 (3H, s), 2.86 (4H, m, overlapping), 2.65 (1H, m), 1.25 (1H, m); $^{13}$C NMR (125 MHz, $D_2O$) δ 178.9, 173.4, 163.6, 146.6, 96.3, 75.3, 68.3, 59.6, 44.7, 42.5, 40.6, 40.1, 30.9; HR–ESI–MS m/z [M–H]$^-$ 375.0989 (calc. 375.0980, $C_{13}H_{19}N_4O_7S_1^-$)[18].

SB-203207 **2** (2.2 mg) was isolated from a 2.0 L culture of the *S. lividans*/sbz 1 and 2 strain. The culture broth was freeze-dried and subjected to ODS open-column chromatography, eluted with $CH_3OH/H_2O$ (50 mM ammonium acetate) with a gradient from 0 to 50% $CH_3OH$. The fractions containing **2** were combined,

freeze-dried, and further purified by semi-preparative HPLC with a Triant C18 column (10.0 mm I.D. × 250 mm, YMC), using 5% $CH_3CN/H_2O$ (50 mM ammonium acetate) as the solvent.

SB-203207 (**2**): white powder; UV ($CH_3OH$) $\lambda_{max}$: 299 nm; $^1$H NMR (500 MHz, $D_2O$) δ 7.26 (1H, s), 4.25 (1H, d, $J$ = 14.2 Hz), 4.15 (1H, t, $J$ = 8.0 Hz), 4.03 (1H, d, $J$ = 14.2 Hz), 3.59 (1H, d, $J$ = 4.3 Hz), 2.86 (3H, s), 2.73 (4H, m, overlapping), 2.54 (1H, m), 1.92 (1H, m), 1.39 (1H, m), 1.13 (2H, m, overlapping), 0.90 (3H, d, $J$ = 7.2 Hz), 0.80 (3H, t, $J$ = 7.5 Hz); $^{13}$C NMR (125 MHz, $D_2O$) δ 175.7, 175.5, 173.4, 164.2, 146.6, 96.2, 75.2, 69.3, 60.3, 57.4, 44.7, 42.6, 40.6, 40.0, 36.3, 30.9, 24.0, 14.6, 11.0. HR–ESI–MS m/z [M–H]$^-$ 488.1819 (calc. 488.1821, $C_{19}H_{30}N_5O_8S_1^-$)[17].

SB-203208 **3** (3.5 mg) was isolated from a 2.0 L culture of the *S. lividans*/sbz 1 and 2 strain. The culture broth was freeze-dried and subjected to ODS open-column chromatography, eluted with $CH_3OH/H_2O$ (50 mM ammonium acetate) with a gradient from 0 to 50% $CH_3OH$. The fractions containing **3** were combined, freeze-dried, and further purified by semi-preparative HPLC with a COSMOSIL Hilic column (10.0 mm I.D. × 250 mm, Nacalai Tesque, Inc.) using 75% $CH_3CN/H_2O$ (50 mM ammonium acetate) as the solvent.

SB-203208 (**3**): white powder; UV ($CH_3OH$) $\lambda_{max}$: 297 nm; [α]30$_D$ = +54.7 (c = 0.43, $H_2O$); $^1$H NMR (500 MHz, $D_2O$) δ 7.25 (6H, m, overlapping), 5.48 (1H, m), 4.16 (1H, d, $J$ = 5.7 Hz), 4.15 (1H, d, $J$ = 14.0 Hz), 4.05 (1H, d, $J$ = 14.0 Hz), 3.56 (1H, d, $J$ = 4.2 Hz), 3.34 (1H, m), 3.05 (1H, dd, $J$ = 12.9 & 5.4 Hz), 2.93 (3H, s), 2.70 (3H, m, overlapping), 2.47 (1H, m), 1.90 (1H, m), 1.35 (3H, d, $J$ = 7.0 Hz), 1.35 (1H, m, overlapping), 1.12 (1H, m), 0.90 (3H, d, $J$ = 7.0 Hz), 0.85 (1H, m), 0.80 (3H, d, $J$ = 7.4 Hz); $^{13}$C NMR (125 MHz, $D_2O$) δ 176.3, 175.7, 173.1, 166.7, 164.2, 146.2, 138.5, 129.3, 128.5, 127.9, 97.6, 78.9, 69.9, 60.2, 58.7, 57.1, 44.3, 42.5, 41.5, 40.6, 39.2, 36.3, 30.3, 24.1, 16.6, 14.6, 11.0; HR–ESI–MS m/z [M–H]$^-$ 649.2666 (calc. 649.2661, $C_{29}H_{41}N_6O_9S_1^-$)[17].

Compound **4**; HR–ESI–MS m/z [M-H]$^-$ 536.1835 (calc. 536.1821, $C_{23}H_{30}N_5O_8S_1^-$).

For DNPH-**7**, the enzyme reaction was performed on a 200 μL scale, consisting of 1 mM L-cysteine, 2 mM TCEP, and 10 μM SbzM, in 50 mM phosphate buffer (pH 8.0) at 30 °C for 1 h. Subsequently, 100 x reactions were set up for structure determination. The mixture was then labeled with 2,4-dinitrophenylhydrazine hydrochloride (TCI) (see below for the labeling method), concentrated, and subjected to semi-preparative HPLC with a Triant C18 column (10.0 mm I.D. × 250 mm, YMC), using 55% $CH_3CN/H_2O$ as the solvents, to afford DNPH-**7** (1 mg). UV ($CH_3CN$) $\lambda_{max}$: 224 nm, 258 nm 359 nm; $^1$H and $^{13}$C NMR see Supplementary Table 2. HR–ESI–MS m/z [M–H]$^-$ 302.0193 (calc. 302.0195).

For DNS-**7b**, the enzyme reaction was performed on a 200 μL scale, consisting of 1 mM L-cysteine, 2 mM TCEP, and 10 μM SbzM in 50 mM Tris-HCl (pH 8.0) buffer, at 30 °C for 1 h. Subsequently, 100 x reactions were set up for structure determination. The mixture was then labeled with dansyl chloride (TCI) (see below for the labeling method), concentrated, and subjected to semi-preparative HPLC with a Triant C18 column (10.0 mm I.D. × 250 mm, YMC), using $CH_3CN$ (0.1% FA, B) / $H_2O$ (0.1% FA, A) as solvents with a gradient from 60% B to 100% B over 15 min, to afford DNS-**7b** (1 mg). UV ($CH_3CN$) $\lambda_{max}$: 207 nm, 255 nm 357 nm; $^1$H and $^{13}$C NMR see Supplementary Table 3. HR–ESI–MS m/z [M+H]$^+$ 357.0561 (calc. 357.0573, $C_{14}H_{17}N_2O_5S_2^+$).

Triacetyl-**3** was prepared by the acetylation of **3**. SB-203208 (**3**, 1 mg) was dissolved in 2 mL of pyridine, and 10 μL of acetyl chloride was added three times every 30 min. The reaction was monitored by LC–MS. The diacetyl-**3** was generated immediately and gradually transformed into the triacetyl-**3**. Once the diacetyl-**3** was completely consumed, 1 mL of water was added. The reaction mixture was freeze-dried and purified by semi-preparative HPLC with a YMC-Triant C18 column (10.0 mm I.D. × 250 mm), using 55% $CH_3OH/H_2O$ (50 mM ammonium acetate) as the solvents. Triacetyl-**3** (0.3 mg): white powder; UV ($CH_3OH$) $\lambda_{max}$: 320 nm; [α]30$_D$ = +5.2 (c = 0.025, $H_2O$); $^1$H NMR (500 MHz, $D_2O$) δ 7.54 (1H, m), 7.14 (5H, m), 5.05 (1H, m), 4.07 (2H, m), 4.03 (1H, d, $J$ = 6.3 Hz), 3.63 (1H, m), 3.12 (5H, m), 2.85 (2H, m), 2.60 (1H, m), 2.48 (1H, m), 2.05 (3H, s), 1.95 (3H, m), 1.90 (3H, m), 1.79 (1H, m), 1.35 (1H, m), 1.19 (1H, m), 1.15 (3H, d, $J$ = 7.3 Hz), 1.07 (1H, m), 0.83 (3H, d, $J$ = 7.0 Hz), 0.75 (3H, t, $J$ = 7.0 Hz). HR–ESI–MS m/z [M–H]$^-$ 775.2979 (calc. 775.2978, $C_{35}H_{47}N_6O_{12}S_1^-$)[17].

**Gene deletion of sbzF, sbzH, sbzJ, sbzM, sbzN, sbzO, sbzP, sbzQ.** In general, sbz1 was divided into two parts to remove the target gene. The first part was amplified and inserted into the pZH1 vector, while the second part was inserted into the pZH2 vector (Supplementary Data set 1). After the sequences were confirmed by gene sequencing, these two vectors were transformed together into the host *Streptomyces lividans* TK21. These gene deletion strains were cultured in A3M medium, along with the *S. lividans* TK21 and *S. lividans*/sbz 1 strains, for 3 days. Subsequently, 1 mL of each culture broth was collected and mixed with the same amount of acetonitrile. After centrifugation and filtration, these samples were analyzed by LC–MS. The LC–MS analysis was performed on a Bruker Compact qTOF mass spectrometer with a Shimadzu Prominence HPLC system, using a HILIC pak VG-50 2D column (2.0 mm I.D. × 150 mm, Shodex, Tokyo, Japan) with a gradient from 80% $CH_3CN–H_2O$ (0.5% $NH_3$) to 10% over 20 min. The flow rate was 0.2 mL/min. The target was monitored by the negative mode.

The *S. lividans*/Δ*sbzM* strain and the *S. lividans*/Δ*sbzJ* strain were each inoculated into 50 mL of A3M medium. After culturing the strains for 2 days, 5 mg of 2-sulfamoylacetic acid (**5**, Enamine) was added to each culture. After an additional 3 days of culture, the broth was centrifuged and mixed with the same

amount of acetonitrile. The samples were subjected to an LC–MS analysis after filtration.

**Chiral HPLC analysis of β-methylphenylalanine.** The chiral HPLC analysis was performed on a Shimadzu Prominence HPLC system equipped with a Chirex phase 3126 (4.6 mm I.D. x 250 mm) column, using 2 mM CuSO$_4$ in 85% H$_2$O/15% CH$_3$CN, at a flow rate of 1.0 mL/min, monitored at 210 nm.

The reaction mixture (50 μL) for the assay of SbzB contained 10 mM MgCl$_2$, 1 mM DTT, 1 mM ATP, 1 mM β-methylphenylalanine, and 20 μM SbzB in Tris-HCl (pH 8.0) buffer, and was incubated at 30 °C for 1 h. The reactions were quenched by the addition of 50 μL of methanol. Precipitated proteins were removed by centrifugation. A 100 μL portion of the supernatant was subjected to the HPLC analysis, as described above.

**Cloning, expression, and purification of recombination proteins.** The *sbzA*, *sbzB*, *sbzC*, *sbzG*, *sbzI*, *sbzJ*, *sbzK*, *sbzL*, *sbzM*, and *Ssp_IleRS* genes were amplified from pZH1-sbz1, pZH2-sbz2, or gDNA of *Streptomyces* sp. NCIMB 40513 using the primer pairs listed in Supplementary Data set 1, and inserted into the pET28a vector at the *Hind*III and *Nde*I sites. The *sbzL* gene was further amplified from pET28a-sbzL and inserted into the pHSA81 vector. The *sbzA*, *sbzB*, *sbzC*, *sbzI*, *sbzJ*, *sbzM*, and *Ssp_IleRS* genes were expressed in Rosetta2 (DE3) cells by induction with 0.1 mM IPTG at 16 °C. The *sbzG* and *sbzK* genes were expressed in BLR cells, along with pACYC-sfp, by induction with 0.1 mM IPTG at 16 °C. pHSA81-*sbzL* was expressed in *S. lividans* TK21 in 500-mL baffled flasks containing 100 mL of YEME medium, consisting of 0.3% yeast extract, 0.3% malt extract, 0.5% peptone (Bacto), 1% glucose, 3.4% sucrose, 5 mM MgCl$_2$, and 1% glycine (pH 7.2), for 3 days at 30 °C and 160 min$^{-1}$. For protein purification, the pellets were resuspended in lysis buffer (50 mM Tris-HCl, pH 8.0, 300 mM NaCl, 10% glycerol, and 5 mM imidazole) and lysed by sonication on ice. Cellular debris was removed by centrifugation (13,000×g, 60 min, 4 °C). The supernatant was loaded onto Ni-NTA agarose resin (Qiagen) in a gravity flow column, which was washed with Buffer A (50 mM Tris-HCl, pH 8.0, 300 mM NaCl, 10% glycerol) with 20 mM imidazole followed by elution with Buffer A containing 500 mM imidazole. The purified proteins were concentrated and buffer exchanged into Buffer A, using Amicon Ultra filters. The purified proteins were flash-frozen in liquid nitrogen and stored at −80 °C.

**The assay of SbzA.** The reaction mixture (50 μL) contained 10 mM MgCl$_2$, 1 mM DTT, 1 mM ATP, 1 mM L-isoleucine, 1 mM altemicidin, 10 μL of S30 Premix Plus, 9 μL of T7 S30 Extract (from the S30 T7 High-Yield Protein Expression System, which provides all of the necessary components for translation, Promega), and 10 μM SbzA in Tris-HCl (pH 8.0) buffer, and was incubated at 37 °C for 1 h with shaking. The reactions were quenched by the addition of 50 μL of methanol. Precipitated proteins were removed by centrifugation. A 20 μL portion of the supernatant was subjected to an HPLC analysis, as described above.

The bulk RNA was isolated from Streptomyces sp. NCIMB 40513 culture incubated in YEME medium (0.3% yeast extract, 0.3% malt extract, 0.5% peptone, 1% glucose, 34% sucrose, 1% glycine, 0.1% MgCl$_2$ 6H$_2$O, pH 7.0) at 30 °C for 3 days with the combination of cell wall breakage by phenol and stepwise precipitation with isopropanol[44]. The reaction mixture (50 μL) contained 10 mM MgCl$_2$, 1 mM DTT, 1 mM ATP, 1 mM L-isoleucine, 0.1 mM **1**, 3.75 mg/mL bulk RNA and 10 μM SbzA and/or Ssp-IleRS in HEPES (pH 8.0) buffer, and was incubated at 30 °C for 3 h. The reactions were quenched by the addition of 50 μL of acetonitrile. Precipitated proteins were removed by centrifugation. A 20 μL portion of the supernatant was subjected to LC–MS analysis. The LC–MS analysis was performed with a HILIC pak VG-50 2D column (2.0 mm I.D. × 150 mm, Shodex), with a gradient from 80% CH$_3$CN–H$_2$O (0.5% NH$_3$) to 10% over 20 min, at a flow rate of 0.2 mL/min.

**The substrate screening of AMP-ligases (SbzB and SbzL).** The purified protein (10 μM) was incubated with 1 mM substrate in 100 μL of buffer, containing 2.8 mM hydroxylamine, 1 mM dithiothreitol (DTT), 0.4 U/mL pyrophosphatase (Sigma), 0.5 mM ATP, 10 mM MgCl$_2$, and 50 mM Tris-HCl (pH 7.5). The reaction was incubated for 30 min at 30 °C. Subsequently, a 50 μL portion of the mixture was quenched by adding 100 μL of the working reagent from the malachite green phosphate assay kit (Enzo). After a 20 min incubation at room temperature, the absorbance at 620 nm was measured. The control A$_{620}$ value was subtracted from the A$_{620}$ value of the reaction mixture, and then the relative adenylation activity was calculated.

**LC–MS analysis of PCPs (SbzG and SbzK).** The analysis was performed on a Bruker Compact qTOF mass spectrometer with a Shimadzu Prominence HPLC system, using a COSMOSIL Protein-R column (2.0 mm I.D. × 150 mm, Nacalai Tesque, Inc.). A gradient elution method was used by employing Solvent A (H$_2$O with 0.1% TFA) for 5 min, followed by a gradient to 100% solvent B (CH$_3$CN with 0.1% TFA) over 20 min, at flow rate of 0.2 mL/min. The mass spectrometric analyses were performed with an electrospray ion source operating in the positive mode. The instrument parameters were as follows: capillary voltage 4500 V; nebulizer gas 3.0 L/min; dry gas 6 L/min; dry temperature 200 °C; funnel 1RF

400 Vpp; funnel 2RF 400 Vpp; hexapole RF 500 Vpp; collision energy 10 eV; collision RF 1000 Vpp.

The samples were prepared from the reactions of SbzB/L and SbzG/K. The reactions (50 μL) contained 10 mM MgCl$_2$, 1 mM DTT, 1 mM ATP, 1 mM β-methylphenylalanine/2-sulfamoylacetic acid, 20 μM SbzB/L, and 250 μM SbzG/K in Tris-HCl (pH 8.0) buffer, and were incubated at 30 °C for 1 h. The reactions were quenched by the addition of 0.1% TFA, followed by centrifugation and filtration. A 20 μL portion of each sample was subjected to the LC–MS analysis, as described above.

**The assay of SbzB, SbzC, and SbzG/K.** The reaction mixture (50 μL) for the assay of SbzB, SbzC, and SbzG/K contained 10 mM MgCl$_2$, 1 mM DTT, 1 mM ATP, 1 mM β-methylphenylalanine (Enamine Ltd.), 1 mM altemicidin (**1**) or SB-203207 (**2**), 20 μM SbzB, 20 μM SbzC, and 250 μM SbzG/K in Tris-HCl (pH 8.0) buffer, and was incubated at 30 °C overnight. The reactions were quenched by the addition of 50 μL of methanol, followed by centrifugation to remove the precipitated proteins. A 20 μL portion of the supernatant was subjected to the HPLC analysis, as described above.

**The assay of SbzL, SbzI, and SbzG/K.** The reaction mixture (50 μL) contained 10 mM MgCl$_2$, 1 mM DTT, 1 mM ATP, 1 mM 2-sulfamoylacetic acid (**6**, Enamine Ltd.), 1 mM **5**, 20 μM SbzL, 20 μM SbzI, and 250 μM SbzG/K in 50 mM Tris-HCl (pH 8.0) buffer, and was incubated at 30 °C overnight. The reactions were quenched by the addition of 50 μL of methanol. The precipitated proteins were removed by centrifugation. A 20 μL portion of the supernatant was subjected to the HPLC analysis, as described above.

**The assay of SbzM and SbzJ.** The reaction mixture (50 μL) contained 2 mM tris (2-carboxyethyl)phosphine (TCEP), 1 mM L-cysteine, 1 mM NAD$^+$, 10 μM SbzM, and 10 μM SbzJ, in 50 mM phosphate buffer (pH 8.0), and was incubated at 30 °C for 30 min.

**The analysis of the labeled compounds.** For carboxy group labeling, 20 μL of the reaction mixture was treated with 20 μL of 100 mM KH$_2$PO$_4$ and 100 μL of 4-bromophenacyl bromide (TCI) solution, and incubated at 77 °C for 60 min.

For amino group labeling, 50 μL of the reaction mixture was treated with 50 μL of 80 mM Li$_2$CO$_3$, 70 μL of CH$_3$CN, and 30 μL of dansyl chloride (TCI), and incubated at 25 °C for 1 h.

For aldehyde labeling, 50 μL of the reaction mixture (reaction in phosphate buffer instead of Tris buffer) was treated with 50 μL of 1 M HCl, 50 μL of CH$_3$OH, and 50 μL of 2,4-dinitrophenylhydrazine hydrochloride (TCI), and incubated at 40 °C for 1 h.

After centrifugation, 20 μL of the supernatant was subjected to an HPLC analysis. The solvent system was H$_2$O with 0.1% formic acid (solvent A) and CH$_3$CN with 0.1% formic acid (solvent B), with a gradient of 5–100% B over 30 min. The target was monitored at 254 nm.

LC–MS analysis of the reactions with labeling was performed with a COSMOSIL 5C18-AR-II column (2.0 mm I.D. × 150 mm, Nacalai Tesque, Inc.) with a gradient from 5% CH$_3$CN–H$_2$O (50 mM ammonium acetate) to 100% over 20 min, at a flow rate of 0.1 mL/min.

**LC–MS analysis of 5 and 7.** The LC–MS analysis of the reactions without labeling (quenched by the addition of 50 μL of acetonitrile) was performed with a HILIC pak VG-50 2D column (2.0 mm I.D. × 150 mm, Shodex) with a gradient from 80% CH$_3$CN–H$_2$O (0.5% NH$_3$) to 10% over 20 min, at a flow rate of 0.2 mL/min. The target was monitored by the negative mode.

**H$_2$$^{18}$O incorporation.** H$_2$$^{18}$O incorporation by SbzM and SbzJ was assessed in a reaction (50 μL), containing 2 mM TCEP, 1 mM L-cysteine, 1 mM NAD$^+$, 10 μM SbzM, and 10 μM SbzJ in 50 mM phosphate buffer (pH 8.0) (43 μL of water), incubated at 30 °C for 1 h. The reactions were quenched by the addition of 50 μL of CH$_3$CN. The LC–MS analysis of the reactions was performed as described above.

H$_2$$^{18}$O incorporation by SbzM was assessed in a reaction (50 μL), containing 2 mM TCEP, 1 mM L-cysteine, 10 μM SbzM, and 50 mM phosphate buffer (pH 8.0) (45.5 μL of water), incubated at 30 °C for 1 h. The reactions were quenched by the addition of 50 μL of acetonitrile. The LC–MS analysis of the reactions was performed with a HILIC pak VG-50 2D column (2.0 mm I.D. × 150 mm, Shodex), with a gradient from 80% CH$_3$CN–H$_2$O (50 mM ammonium acetate, pH 8.0) to 10% over 20 min, at a flow rate of 0.2 mL/min. The target was monitored by the negative mode.

**Kinetic analysis of SbzM.** To determine the kinetic parameters of SbzM, the temperature, pH, and time course of the enzymatic assay were monitored. The optimized assays contained the substrate, L-cysteine (0.1, 0.2, 0.4, 0.6, 0.8 mM, in duplicate), and purified SbzM (6 μM). The substrate was mixed with TCEP for 30 min before the assay. The enzyme, in 50 mM Tris buffer (pH 9.0), was pre-incubated at 30 °C for 3 min. A 1 μL portion of each substrate mixture was added and further incubated for 2 min at 30 °C. The enzymatic reactions were stopped by the addition of 50 μL of acetonitrile. Consumption of the substrate was quantified

by an LC–MS analysis. The kinetics values were calculated with GraphPad Prism 6.0 (San Diego, California, USA). The LC–MS analysis was conducted with a Bruker Compact qTOF mass spectrometer with a Shimadzu Prominence HPLC system, using a HILIC pak VG-50 2D column (2.0 mm I.D. × 150 mm, Shodex) with a gradient from 80% $CH_3CN/H_2O$ (20 mM formic acid) to 10% over 15 min. The chromatograms were extracted at an m/z 122.0270 ± 0.01 ([M+H]$^+$ for L-cysteine).

**Metal chelation and reconstitution of SbzM.** The purified SbzM enzyme was incubated with 1 mM EDTA at 30 °C for 1 h. The chelation of the active-site metal was verified by the loss of enzyme activity (dansyl chloride method as described above). The excess EDTA was subsequently removed by dialysis at 4 °C overnight. Different metal ions ($Fe^{2+}$, $Cu^{2+}$, $Zn^{2+}$, $Mn^{2+}$, $Ni^{2+}$, $Co^{2+}$) were supplemented to the dialyzed apo-protein at a 2 mM concentration and incubated at 30 °C for 1 h. The recovery of the enzyme activity was subsequently measured.

**ICP-MS analysis of SbzM.** The metal contents of SbzM were determined by inductive-coupled plasma mass spectrometry (ICP-MS) using a Thermo iCAP-Q system under standard operating conditions. The protein samples were prepared as follows: the initial protein solution (10 μM) was taken as a positive control; the protein solution after 1 mM EDTA treatment and twice dialysis was taken as a negative control; then the protein solutions after 2 mM Fe, Cu, and Ni treatment and followed by twice dialysis were taken as Fe, Cu, and Ni, respectively (final metal concentration in the solution is 0.014 μM). This analysis indicated that the content of Fe and Ni increase significantly while no increase observed for other metals, respectively.

**Hydrolysis of 1 to prepare 6.** Altemicidin **1** (2 mg) was hydrolyzed with 4 M HCl (1.5 mL) at 50 °C overnight. The reaction mixture was freeze-dried and purified by chromatography on a Cosmosil HILIC column (10 mm I.D. × 250 mm, Nacalai Tesque, Inc.; flow rate 3 mL/min; 70% $CH_3CN/H_2O$ containing 50 mM ammonium acetate). **6**: white powder; UV ($CH_3OH$) $\lambda_{max}$: 294 nm; HR–ESI–MS m/z [M–H]$^-$ 254.1148 (calc. 254.1146, $C_{11}H_{16}N_3O_4^-$).

**Isotope-labeled precursor feeding.** L-cysteine-$^{13}C_3$, $^{15}N$ (100 mg) was fed to the altemicidin-producing strain (S. lividans/sbz 1) after 48 h, 60 h, and 72 h incubations in 100 mL of Hijacking medium (0.3% soy broth, 0.5% pharma media, 0.3% yeast extract, 2.0% black treacle, 2.0% glucose, and 0.4% $CaCO_3$). After 6 days of fermentation, the labeled **1** was isolated as described above.

**Reporting Summary.** Further information on experimental design is available in the Nature Research Reporting Summary linked to this article.

## Data availability

The DNA sequences of altemicidin, SB-203207, and SB-203208 biosynthetic gene clusters (sbz cluster) and Ile-tRNA synthetase gene from Streptomyces sp. NCIMB40513 (Ssp-IleRS) were registered as LC420052 and LC420053 in GenBank, respectively. The data that support the findings are available on request. A reporting summary for this article is available as a Supplementary Information file.

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

## Acknowledgements

This work was supported by a Grant-in-Aid for Scientific Research from the Ministry of Education, Culture, Sports, Science and Technology, Japan (JSPS KAKENHI grant numbers JP16H06443, JP17H04763, and JP18K19139), JST/NSFC Strategic International Collaborative Research Program Japan–China, and Kobayashi International Scholarship Foundation. We appreciate GlaxoSmithKline for providing *Streptomyces* sp. NCIMB 40513 and Institute of Microbial Chemistry for providing the authentic sample of altemicidin. We also thank Dr. Kazuo Furihata for discussion on NMR, Dr. Hiroyasu Onaka for providing pTYM19ep, Dr. Michihiko Kobayashi for providing pHSA81, and Dr. Toshiyuki Kan for providing the authentic sample of SB-203207.

## Author contributions

T.A. and I.A. designed the experiments. Z.H. and T.A. performed the experiments. Z.H., T.A., Z.M., and I.A. analyzed the data. Z.H., T.A., and I.A. wrote the paper.

## Additional information

**Competing interests:** The authors declare no competing interests.

