## [Peer Review File · Nature Communications]

Reviewers' comments:

Reviewer #1 (Remarks to the Author):

Hu et al report a very nice study on the biosynthesis of a sulfonamide-bearing natural product and present a novel route to this pharmacologically important moiety. Specifically, they identified the gene cluster coding for the antibiotics altemicidin and two congeners from a *Streptomyces* species and investigated the pathway by heterologous gene expression and biochemical analyses. They discovered novel aminoacyl transfer reactions, including the tRNA synthetase-like and GNAT-like enzymes as well as a cupin dioxygenase, which catalyzes a remarkable oxidation/decarboxylation of cysteine and the subsequent intramolecular amino group rearrangement yielding an N-S bond. The experimental design and the results are very convincing. The data are also nicely presented. In light of the significance of the pathway I think Nature Communications is the right journal to publish this interesting finding after only minor changes:

- 1) This is not the first reported enzymatic sulfonamide formation as stated in the abstract and elsewhere. Claims like "naturally occurring sulfonamides are rare, and their biosynthesis has not been investigated" and "this is the first report of the unprecedented biosynthetic machineries of aminoacyl sulfonamide antibiotics" need to be toned down. Some sulfadixiamycins are naturally occurring sulfonamides, and their biosynthesis has been studied in the Hertweck lab. Likewise, the biosynthesis of ascamycin/dealanylascamycin has been reported, as stated by the authors (see refs 11 and 12). This study is sufficiently intriguing and relevant even though it is not the first study on sulfonamide biosynthesis. Please rephrase (and perhaps add qualifiers such as "aliphatic" sulfonamides?)
- 2) In Figure 4b/c the step from 6b to 6 is quite complex. Are there any precedents from synthesis (literature?) or is there other evidence.
- 3) from *Streptomyces*: species name is missing; from a *Streptomyces* sp. or: from a streptomycete
- 4) References: several journal abbreviations need to be fixed

Reviewer #2 (Remarks to the Author):

The manuscript by Hu et al reports the identification of a new biosynthetic gene cluster from *Streptomyces* that is responsible for producing sulfonamide antibiotics. The work is novel and notable for its identification of a number of unique and interesting enzymes that are involved in this process, and will be of wide interest and perhaps have substantial impact in the field. Although much of the work is sound and the basic identification of the gene cluster is clearly established, there are a number of important details that are not sufficiently well established, especially with respect to the IleRS-like enzymes and reactions. Additional data is necessary before the manuscript could be suitable for publication in *Nature Communications*.

The analysis of IleRS structure-function relationships is incomplete and may be flawed. Both genes encoding IleRS homologs have similar sequence identity to *E. coli* IleRS, so it is not immediately evident which gene might code for IleRS and which for sulfonamide biosynthesis. The authors choose *sbzA* for the latter function clearly because it is present in the biosynthetic operon. However, the sequence alignment does not support the assignment of the other *Ssp_IleRS* as an active IleRS. One of the two signature ATP binding motifs in the Rossmann fold of class I aaRS is HIGH, but only the Gly in this sequence is fully conserved and essential. It forms a platform for stacking of the adenine ring of ATP, and any beta-methyl at this position sterically excludes proper ATP binding. The problem is that *Ssp_IleRS* has a His at this position (Supp Fig 32) and is predicted to be inactive in aminoacylation of tRNA-Ile for this reason, while *SbzA* has the Gly residue. Is there a mistake here in the sequence, or are there novel compensating factors that allow *Ssp_IleRS* to function as an IleRS despite the lack of the crucial Gly? Does *SbzA* perhaps perform both functions? Does the other class I aaRS ATP binding motif, the KMSKS sequence, conform to expectations? Also, it would be good to see an analysis of the Ile binding site in the two proteins - it is poorly shown in Supp Fig 32b.

Related to these issues is that only a single paper from the field of aminoacyl-tRNA synthetases is cited in the manuscript (ref 33 and again as ref 4 of the supplement). This reinforces my concern that the authors have not made an in-depth study of structure-function relationships, as relevant to the unusual presence of two *ilers* genes in this organism. In the discussion, there is only a single sentence stating that *SbzA* does "not completely conserve" the ATP binding site. I am an expert on aaRS, yet I see no obvious justification in the alignment for this statement. What exactly is not conserved? The authors should provide full details and additional figures to justify this statement and to resolve the ambiguity about the Gly residue described above.

The experiments to establish the activity of *SbzA* as an aminoacylating enzyme are incomplete and require further work. The authors provide Ile-tRNA(Ile) by adding in an aliquot of an *E. coli* cell lysate. They show a peak coinciding in mobility with product 2 that is RNase-sensitive (Fig 3a), and this is consistent with but does not "confirm" that Ile-tRNA(Ile) is involved, as stated in the text (p 5). Also, the peak in Fig 3a is quite small, possibly because the authors chose to use a heterologous *E. coli* extract with an evolutionarily divergent protein translation apparatus. The experiments should

be repeated with bulk tRNA and with Ile-tRNA isolated from a *Streptomyces* extract. Ile-tRNA can be easily purified by oligo-affinity chromatography and aminoacylated in vitro by IleRS. Conducting the assay with pure (or at least bulk) tRNA and, further, doing so in parallel with the protein products of both *sbzA* and *Ssp_IleRS* genes, would strengthen the findings of the paper and definitively establish the functions of the two genes. This is of particular importance given the ambiguity in the sequence alignment.

Other points -

-In Figure 2 there is a peak in sample (i), the control, which appears to exactly coincide in mobility with the peak for compound 2. That peak is absent in sample (ii) as would be expected if *sbz2* is required for the biosynthesis of compound 2. What is this peak in the background extract, which appears to compromise the interpretation of this experiment?

-For the experiments on the adenylate forming enzyme *SbzB*, described in the bottom paragraph of p 5, the identity of the expected product is not clearly stated, and the primary data (with appropriate controls) are not included in the supplementary figures 11 and 9 that are cited.

-Figure 1 should include drawings of compounds 4 and 5 that are included in Fig 3d. Compound 4 appears not to be depicted anywhere.

Reviewer #3 (Remarks to the Author):

In this paper, the authors investigated the biosynthetic basis for formation of a sulfonamide in a natural products, which has not been reported previously. They focused on a Ile-tRNA synthetase inhibitor, which allowed them to look for a putative additional copy of Ile-RS on the genome of the producer organism. This additional copy would then likely be a resistant mutant that protects the producing organism. This clever approach worked and the authors found a candidate cluster.

They then used heterologous production to verify that two operons together indeed make the desired compound. Surprisingly, the IleRS-like enzyme *SbzA* is not just a resistant copy but actually

participates in the biosynthesis. It adds Ile to a sulfonamide group to make a isoleucylsulfonamide in vitro when provided with Ile-tRNA. The authors then investigated the other amide forming enzymes in the cluster. They show that SbzC, a member of the GNAT family, forms the CoA thioester of one stereoisomer of beta-methyl-Phe, but that the resulting CoA thioester is not incorporated directly into the biosynthetic pathway. Instead the CoA thioester is first loaded onto a carrier protein (both SbzG and SbzK work in this step) and then transferred to 2. This reaction does not appear to work very well based on the peak size of the product in Fig. 3b. The authors also assign function for SbzL, although again the activity in Fig 3c is not very robust.

The low activity of SbzC in Fig 3b could be because the enzyme is not well behaved in vitro. But it is also possible that perhaps these are not the correct substrates. Did the authors try if this reaction works with compounds 1 or 5 as acceptor of the methylphenylalanine? In turn the low activity seen in the other panels of Fig 3 could then be because the order of the reactions is not that shown in Fig 3d.

The most interesting part of the paper is the production of sulfonamide 4 from Cys. Gene deletion studies implicated SbzM and SbzJ in its biosynthesis, which are a cupin dioxygenase and aldehyde dehydrogenase, respectively. The authors show convincingly that Cys is converted to hydrated sulfonamide aldehyde 6, which is a remarkable reaction. They propose that SbzJ would then oxidize the aldehyde to the carboxylic acid. The authors suggest that the SbzM activity can be achieved with Ni²⁺, Cu²⁺ or Fe²⁺ but I find that unlikely since the reactivity of these metals is very different. The authors should separate the reconstituted enzyme from the metal solution and then determine by IPC-MS what metal is really bound. Because at 2 mM metal (enzyme concentration is not given in Supp Info) 2% impurity could fully reconstitute 20 uM enzyme.

The reaction catalyzed by SbzM is very interesting, but the proposed mechanism in Fig 4d and 4e must be revised since it has several steps that chemically are highly unlikely as drawn. In panel d, the proton that is deprotonated by the base B⁻ is not acidic. In addition, the electrons are pushed towards the thiol which is not electrophilic, and pushing electrons towards a thiol does not lead to a thioaldehyde (you need to remove electrons from the thiol, not push them towards the thiol). Instead, almost certainly with a cupin (and in my opinion very likely with Fe²⁺), the ferrous enzyme will activate O₂ and the activated enzyme will abstract a hydrogen atom to oxidize the thiol to the thioaldehyde (not unlike what the authors draw in SI Fig 33). I note that the authors also write the thioaldehyde as the product but the arrows would NOT provide an aldehyde since no electrons are removed (a deprotonation is not an oxidation). There is a similar problem with panel e. The arrows in 6b push electrons to the amine that is not electrophilic and the results of the arrow pushing is NOT 6c. Compound 6b has a negative charge whereas 6c is not charged. Those electrons must go somewhere. I strongly encourage the authors to revisit the mechanism since as drawn it is incorrect.

In summary, I think this paper reports two very interesting pieces of new data. First the assumed resistance protein actually participates in the biosynthesis (I am not aware of other examples). Second SbzM catalyzes a fascinating reaction that provides the first insights into sulfonamide formation. If the authors can address the questions raised above, I think this study could be a good fit for Nat Commun.

Minor

p. 7: “were 3 larger than the original one” It would be clearer to say “were 3 Da larger than the original one

Fig S33. Please add metal oxidation states. Also is it known that the oxidation requires a single molecule of O₂ as drawn? Or could it involve two molecules of O₂ in two consecutive oxidations?

p. 8. Please correct sulfone amide. These compounds do not contain a sulfone.

p. 9. “The aminoacyl moieties introduced by the known aa-tRNA-dependent transferases in secondary metabolism are Ser, Leu, Gly, and Ala,”. Don’t the cyclodipeptide synthetases use aromatic amino acids in natural product biosynthesis?

THE UNIVERSITY OF TOKYO

We have addressed the substantive points and technical details as outlined below (changes highlighted in red in the text).

Reviewer #1

1. *This is not the first reported enzymatic sulfonamide formation as stated in the abstract and elsewhere. Claims like "naturally occurring sulfonamides are rare, and their biosynthesis has not been investigated" and "this is the first report of the unprecedented biosynthetic machineries of aminoacyl sulfonamide antibiotics" need to be toned down. Some sulfadixiamycins are naturally occurring sulfonamides, and their biosynthesis has been studied in the Hertweck lab. Likewise, the biosynthesis of ascamycin/dealanylascamycin has been reported, as stated by the authors (see refs 11 and 12). This study is sufficiently intriguing and relevant even though it is not the first study on sulfonamide biosynthesis. Please rephrase (and perhaps add qualifiers such as "aliphatic" sulfonamides?)*

Many thanks for the comment. According to the suggestion, we revised the text in abstract.

2. *In Figure 4b/c the step from 6b to 6 is quite complex. Are there any precedents from synthesis (literature?) or is there other evidence.*

We revised the reaction scheme of SbzM significantly. Please see our response below to Reviewer #3.

3. *from Streptomyces: species name is missing; from a Streptomyces sp. or: from a streptomycete*

According to the suggestion, we revised the text in abstract.

4. *References: several journal abbreviations need to be fixed*

Done.

Reviewer #2

Major comments #1

1. *The analysis of IleRS structure-function relationships is incomplete and may be flawed. Both genes encoding IleRS homologs have similar sequence identity to E. coli IleRS, so it is not immediately evident which gene might code for IleRS and which for sulfonamide biosynthesis. The authors choose sbzA for the latter function clearly because it is present in the biosynthetic operon. However, the sequence alignment does not support the assignment of the other Ssp_IleRS as an active IleRS. One of the two signature ATP binding motifs in the Rossmann fold of class I aaRS is HIGH, but only the Gly in this sequence is fully conserved and essential. It forms a platform for stacking of the adenine ring of ATP, and any beta-methyl at this position sterically excludes proper ATP binding. The problem is that Ssp_IleRS has a His at this position (Supp Fig 32) and is predicted to be inactive in aminoacylation of tRNA-Ile for this reason, while SbzA has the Gly residue. Is there a mistake here in the sequence, or are there novel compensating factors that allow Ssp_IleRS to function as an IleRS despite the lack of the crucial Gly?*

We appreciate the thoughtful comment. We carefully checked the DNA sequence of Ssp_IleRS to confirm it is correct. Notably, it has been also reported in the literature that the "HIGH" motif of IleRSs from *Streptomyces coelicolor* and *Streptomyces griseus* (SclIleRS and SglIleRS), both possessing the aminoacylation activities, was replaced with "GAHH" (Cvetesic, N. *et al.*, *J. Biol. Chem.* 291, 8618, 2016). Future structural study is prerequisite to understand how these IleRS enzymes bind the ATP molecule without this crucial Gly residue. Please also see our response below.

Does SbzA perhaps perform both functions?

We carefully repeated the experiment and indeed detected the aminoacylation activity of SbzA in addition to the aminoacyltransfer from Ile-tRNA. However, it is much weaker than that of Ssp_IleRS. Please also see the response below.

Does the other class I aaRS ATP binding motif, the KMSKS sequence, conform to expectations? Also, it would be good to see an analysis of the Ile binding site in the two proteins - it is poorly shown in Supp Fig 32b.

Ssp_IleRS, Sc_IleRS, and Sg_IleRS conserve the “KMSKH” sequence as an ATP binding motif, while in the case of SbzA, it is replaced with “AMSKA”. We also updated the figure (Supp Fig 34b).

2. Related to these issues is that only a single paper from the field of aminoacyl-tRNA synthetases is cited in the manuscript (ref 33 and again as ref 4 of the supplement). This reinforces my concern that the authors have not made an in-depth study of structure-function relationships, as relevant to the unusual presence of two ilers genes in this organism. In the discussion, there is only a single sentence stating that SbzA does "not completely conserve" the ATP binding site. I am an expert on aaRS, yet I see no obvious justification in the alignment for this statement. What exactly is not conserved? The authors should provide full details and additional figures to justify this statement and to resolve the ambiguity about the Gly residue described above.

As mentioned above, several *Streptomyces* aminoacyl-tRNA synthetases including Ssp_IleRS lack the crucial Gly residue but still active. Without structural information of the *Streptomyces* IleRS enzymes, it is hard to understand how these enzymes bind the ATP molecule without the crucial Gly residue. Only from the sequence comparison, we cannot discuss a lot in detail. Although we originally wanted to include the in-depth study of structure-function relationships, this has in fact become two new separate projects and thus constitute a totally separate undertaking. We fully expect to complete these studies, but we feel that the present result can stand alone in terms of substance and novelty as a communication of an important discovery.

We, therefore, in the revised manuscript, only referred to the “HIGH” and “KMSKH” motifs in SbzA, and avoid further detailed discussion on the structure-function relationship. We also modified the references.

3. The experiments to establish the activity of SbzA as an aminoacylating enzyme are incomplete and require further work. The authors provide Ile-tRNA(Ile) by adding in an aliquot of an E. coli cell lysate. They show a peak coinciding in mobility with product 2 that is RNase-sensitive (Fig 3a), and this is consistent with but does not "confirm" that Ile-tRNA(Ile) is involved, as stated in the text (p 5). Also, the peak in Fig 3a is quite small, possibly because the authors chose to use a heterologous E coli extract with an evolutionarily divergent protein translation apparatus. The experiments should be repeated with bulk tRNA and with Ile-tRNA isolated from a Streptomyces extract. Ile-tRNA can be easily purified by oligo-affinity chromatography and aminoacylated in vitro by IleRS. Conducting the assay with pure (or at least bulk) tRNA and, further, doing so in parallel with the protein products of both sbzA and Ssp_IleRS genes, would strengthen the findings of the paper and definitively establish the functions of the two genes. This is of particular importance given the ambiguity in the sequence alignment.

Thank you very much for the thoughtful comment again. According to the suggestion, we newly conducted further experiments. We isolated the bulk tRNA from *Streptomyces* sp. NCIMB40513. The incubation of SbzA and Ssp_IleRS in presence of altemicidin (**1**), the tRNA, and isoleucine successfully yielded SB-203207 (**2**) as a product, suggesting that Ssp_IleRS aminoacylates tRNA^{Ile} and SbzA transfers isoleucyl moiety on **1**. Further, to our surprise, the SbzA reaction without Ssp_IleRS yielded a trace amount of **2**, implying that SbzA also can produce Isoleucyl-tRNA. Given that the amount of **2** from SbzA sole reaction is 94% less than that of SbzA+Ssp_IleRS reaction, we concluded that Ssp_IleRS is mainly responsible for tRNA aminoacylation, and SbzA is responsible for the aminoacyl transfer from Ile-tRNA onto **1** to yield **2**.

Based on these newly obtained experimental results, we added the following sentences and a figure in the result section (Page 5).

“We also conducted the SbzA enzyme reaction with **1** and Ile-tRNA_{Ssp}^{Ile} which was synthesized by Ssp-IleRS from bulk tRNA isolated from *Sreptomycetes* sp. NCIMB40513, and detected **2** as a product (**Supplementary Fig. 12**). This result reconfirmed the dependency of SbzA reaction on Ile-tRNA. Interestingly, SbzA alone can also produce a trace amount of **2** in the absence of Ssp-IleRS, even though the yield is 94% less than SbzA+Ssp_IleRS reaction. This data suggested that SbzA can also catalyze the isoleucyl transfer reaction onto tRNA^{Ile}, but with much less efficiency than that of Ssp_IleRS.”

We also added the sentences in the discussion section (Page 9-10) as below.

“The *in vitro* assay of SbzA with bulk tRNA showed that Ssp_IleRS aminoacylates tRNA^{Ile}, and then, SbzA installs isoleucyl moiety on **1**. Thus, we concluded that Ssp_IleRS catalyzes the tRNA aminoacylation, and SbzA in the gene cluster catalyzes the aminoacyl transfer from Ile-tRNA onto **1** to yield **2**. Interestingly, SbzA can also catalyzes aminoacylation of tRNA^{Ile}, but with much less efficiency than that of Ssp_IleRS. Notably, SbzA possesses conserved “HIGH” and “KMSKS” sequences which are signature ATP binding motifs in the Rossmann fold of class I aaRS³⁴, even though there are slight modification (**Supplementary Fig. 34**), which could support the observation that SbzA also catalyzes the tRNA acylation. The detailed structure-function relationship on SbzA should be the subject of our future study.”

Other points

-In Figure 2 there is a peak in sample (i), the control, which appears to exactly coincide in mobility with the peak for compound 2. That peak is absent in sample (ii) as would be expected if sbz2 is required for the biosynthesis of compound 2. What is this peak in the background extract, which appears to compromise the interpretation of this experiment?

Because the UV_{max} of the peak eluted at the same RT as **2** is 275 nm, we think this compound is not relevant to **1-3** biosyntheses.

-For the experiments on the adenylate forming enzyme SbzB, described in the bottom paragraph of p 5, the identity of the expected product is not clearly stated, and the primary data (with appropriate controls) are not included in the supplementary figures 11 and 9 that are cited.

We revised the text as follows. “However, the reaction did not yield the expected product **3** (**Fig. 3b**)” (Page 6). We omitted the primary data of Supplementary figures 11 and 9, as in the case of other previous published reports for AMP-ligase assays, since it is just the enumerate of number.

-Figure 1 should include drawings of compounds 4 and 5 that are included in Fig 3d. Compound 4 appears not to be depicted anywhere.

We would like to keep Figure 1 as the current version so that it shows up the target antibiotics in this study. Instead, we newly added the structure of 2-sulfamoylacetic acid (**5**) (originally labeled as Compound **4**) into Figure 3d. The structure of compound **6** (originally labeled as Compound **5**) was originally included in Figure 3d.

Reviewer #3

1. The low activity of SbzC in Fig 3b could be because the enzyme is not well behaved in vitro. But it is also possible that perhaps these are not the correct substrates. Did the authors try if this reaction works with compounds 1 or 5 as acceptor of the methylphenylalanine? In turn the low activity seen in the other panels of Fig 3 could then be because the order of the reactions is not that shown in Fig 3d.

Thank you very much for the thoughtful comment. According to the suggestion, we repeated the experiment by employing **1** as a substrate for the SbzC enzyme reaction. As a result, we found that the yield of the methylphenylalanylated product (now newly labeled as **4**) is lower than that of **3** from the reaction using **2** as a substrate. This result suggests that the order of the reaction in Figure 3d is correct.

Based on these newly obtained experimental results, we added the following sentences and a figure in the result section (Page 6).

“We also tested **1** as an acceptor substrate for SbzBCG reaction, and the detected **4** as a product, but with much less efficiency (**Supplementary Fig. 14**).”

2. *The authors suggest that the SbzM activity can be achieved with Ni²⁺, Cu²⁺ or Fe²⁺ but I find that unlikely since the reactivity of these metals is very different. The authors should separate the reconstituted enzyme from the metal solution and then determine by IPC-MS what metal is really bound. Because at 2 mM metal (enzyme concentration is not given in Supp Info) 2% impurity could fully reconstitute 20 uM enzyme.*

We appreciate this thoughtful and very important comment. According to the suggestion, we newly conducted experiments. Through ICP-MS analyses, we confirmed that the content of Fe²⁺ and Ni²⁺ increase while no increase observed for other metals. Based on this result, we concluded that Fe²⁺ and Ni²⁺ really bind to the enzyme, and they are important for the activity. On the other hand, the enzyme incubated with Cu²⁺ precipitated after the successive dialysis to reduce the concentration of each metal to significantly lower than the enzyme concentration.

Based on these newly obtained experimental results, we newly added the following sentences and a figure (Page 8-9).

“Furthermore, we confirmed that the contents of Ni²⁺ and Fe²⁺ increased while no increase for other metal ions in each of the enzymes by ICP-MS analyses. Therefore, we concluded that Fe²⁺ and Ni²⁺ bind to the enzyme and are essential for the SbzM enzyme reaction. On the other hand, the enzyme incubated with Cu²⁺ precipitated after the successive dialysis to reduce the concentration of each metal to significantly lower than the enzyme concentration.”

We also added the methods for ICP-MS analysis in Page 22 accordingly.

3. *The reaction catalyzed by SbzM is very interesting, but the proposed mechanism in Fig 4d and 4e must be revised since it has several steps that chemically are highly unlikely as drawn. ... I strongly encourage the authors to revisit the mechanism since as drawn it is incorrect.*

Thank you very much for the very thoughtful comment again. We revised the reaction scheme significantly. Yes, we agree the previous reaction scheme, which includes the sulfene formation from sulfonate, is not reasonable. We also noticed that it is difficult to propose the oxidation step from sulfinic to sulfonate. Thus, we newly propose the S-N bond formation from the intermediate containing a sulfonate-enamine structure.

We added the following sentences and a figure (Page 9-10 and Fig. 5).

“The reaction from L-cysteine to **7a** was hypothesized as below (**Fig. 5a**). The cysteine binding to the enzyme changes the coordination of the iron, followed by the binding of the molecular oxygen to the ferrous ion and successive nucleophilic addition to yield **7a-1**, in a similar manner to cysteine dioxygenase³⁷. The resultant oxide anion abstracts the C β proton and the carbon-sulfur double bond is newly formed in **7a-2**. This C β proton might be supported through ionic interaction with the amino acid residues in SbzM as seen in EpiD, an flavin-dependent cysteine decarboxylase³⁸. An evolving negative charge on sulfur moves to oxygen, resulting in **7a-3** sulfene structure which is generated through elimination of HCl from sulfinyl chloride in chemical synthesis³⁹. Finally, this intermediate generates a more stable ene-thioaldehyde S-oxide⁴⁰ **7a**.

The oxidation mechanism of **7a** to **7b** is predicted to resemble that proposed based on the the X-ray structure of the cupin-type cysteine dioxygenase complexed with cysteine persulfenate³⁷ (**Fig. 5b**). The ferrous-bound molecular oxygen is attacked by the electron from the sulfanolate in nucleophilic addition mechanism to give **7b-1**. The addition of the negatively-charged distal oxygen to the sulfoxide generates thiadioxirane **7b-2**. The migration of the lone pair on the sulfur driven by the electron migration from the sulfanolate and the cleavage of the O-O bond yield **7b**. However, we cannot exclude the possibility of the S-O bond formation through radical mechanism in this step as discussed in the literature³⁷. Finally, **7b** tautomerizes to **7c**, the resultant imine of **7c** is hydrolyzed, and the hydrolyzed amine in **7d** attacks the protonated sulfonic acid, resulting in **5** accompanied with elimination of water as a leaving group (**Fig. 5c**).”

We also revised the text as follows.

"Taken together, for the biosynthesis of the sulfonamide **5**, we propose that the cupin dioxygenase SbzM catalyzes the decarboxylation coupled with monooxygenation to yield (Z)-(2-aminovinyl)sulfanolate **7a** and dioxygenation of L-cysteine to yield the reactive aminosulfonate **7b**." (Page 8)

"In the biogenesis of a sulfonamide, we propose that the cupin dioxygenase SbzM catalyzes the sequential two-step oxidations of L-cysteine to yield the reactive (Z)-2-aminoethanone-1-sulfonate (**7b**) via (Z)-(2-aminovinyl)sulfanolate (**7a**), and the subsequent intramolecular rearrangement of the amino group onto the sulfur atom (N-S bond formation) generates the sulfonamide aldehyde **7** (**Fig. 4d**)." (Page 10)

"SbzM is the first cupin-type cysteine decarboxylase. Oxalate decarboxylase is a cupin-type decarboxylase, but its reaction mechanism is likely to be different from the one that we proposed⁴³. The structural studies on SbzM is now in progress to further clarify its reaction mechanism." (Page 11)

"In addition, homologs of the cupin dioxygenase SbzM are encoded in the genomes of several bacteria, and interestingly, some of them are clustered with secondary metabolism-like genes (**Supplementary Fig. 35**), suggesting the distribution of the SbzM-catalyzed cysteine metabolism that generates the reactive sulfonate-enamine compounds." (Page 12)

Minor

p. 7: "were 3 larger than the original one" It would be clearer to say "were 3 Da larger than the original one"

According to the suggestion, we revised the text.

Fig S33. Please add metal oxidation states. Also is it known that the oxidation requires a single molecule of O2 as drawn? Or could it involve two molecules of O2 in two consecutive oxidations?

We added metal oxidation state and put the scheme into the newly-added Fig. 5. Now, we propose that SbzM' reaction involves the decarboxylation coupled with sulfur monooxygenation and the successive dioxygenation. Please also see our response above.

p. 8. Please correct sulfone amide. These compounds do not contain a sulfone.

According to the suggestion, we revised the text.

p. 9. "The aminoacyl moieties introduced by the known aatRNA-dependent transferases in secondary metabolism are Ser, Leu, Gly, and Ala,". Don't the cyclodipeptide synthetases use aromatic amino acids in natural product biosynthesis?

According to the suggestion, we added the amino acids incorporated by cyclodipeptide synthetase as an example for ones incorporated by aatRNA-dependent transferases. Further, we modified the text as follows.

"The aminoacyl moieties introduced by the known aa-tRNA-dependent transferases in secondary metabolism are Ser, Leu, Gly, Ala, Phe, Tyr, and Trp, and this is the first report of the installation of Ile by a tRNA-dependent aminoacyl transferase³⁵." (Page 10)

We hope you will agree that the manuscript has been significantly improved, and that you will find it acceptable for publication.

Yours sincerely,

Ikuro Abe, Ph.D.

Reviewer #1 (Remarks to the Author):

The authors have adequately revised the manuscript according to my suggestions.

Reviewer #2 (Remarks to the Author):

The revised version of the paper adequately addresses my concerns about the original submission, and can be published in Nature Communications in its present version.

Reviewer #3 (Remarks to the Author):

The authors have improved the manuscript by trying the potential alternative substrates for SbcZ and conclude that the originally suggested substrate gives the best result. As I mentioned earlier, the product peaks with the original substrates are very small, suggesting that something might be missing or not physiological, but I cannot think of other possible experiments. An acknowledgment in the text that the activities appear weak might be appropriate.

The authors also carried out the suggested experiments to make sure that the metals investigated are actually bound to SbzM. Although it is still puzzling to me that the enzyme can work with Fe and Ni, given their very different properties, new Ni activities have been reported in recent years and perhaps this is one of them.

The authors also revised Fig 5, the proposed mechanism of SbzM. It is certainly better than the previous mechanism, although I am not convinced that this mechanism is correct. But the authors refer to literature precedent for some of the steps, and they are free to speculate for the steps that are unprecedented. I do have one more question and that is whether the ¹⁵N from Cys (line 182) does indeed end up in the sulfonamide in 7 at the suggested position (in other words does ¹⁵N NMR agree with this assignment?). I understand that the MS says that the ¹⁵N is incorporated in the product, but I wonder how sure the authors are that SbzM makes the sulfonamide.

The authors have quite a few typo's and grammatical errors in the new text highlighted in red. Here are just a few:

Line 29 knowledges should be singular

Line 37 analyses should be analysis

Line 119 ezyme

Line 120 sreptomycies

Line 211 the authors only are talking about Cu in this sentence, and therefore I do not understand what is meant by "each metal"

Reviewer #3

The authors have improved the manuscript by trying the potential alternative substrates for SbcZ and conclude that the originally suggested substrate gives the best result. As I mentioned earlier, the product peaks with the original substrates are very small, suggesting that something might be missing or not physiological, but I cannot think of other possible experiments. An acknowledgment in the text that the activities appear weak might be appropriate.

According to the suggestion, we modified the text as below.

Page 6

“Interestingly, SbzB accepted both SbzG and SbzK with almost the same preference *in vitro*, and the subsequent O-acyltransfer reaction by SbzC successfully yielded **the small amount of final product 3.**”

The authors also carried out the suggested experiments to make sure that the metals investigated are actually bound to SbzM. Although it is still puzzling to me that the enzyme can work with Fe and Ni, given their very different properties, new Ni activities have been reported in recent years and perhaps this is one of them.

We appreciate the reviewer's thoughtful comment. This point would be answered in our future studies, such as the currently-ongoing X-ray structural studies.

I do have one more question and that is whether the 15N from Cys (line 182) does indeed end up in the sulfonamide in 7 at the suggested position (in other words does 15N NMR agree with this assignment?). I understand that the MS says that the 15N is incorporated in the product, but I wonder how sure the authors are that SbzM makes the sulfonamide.

We failed to detect any ¹⁵N NMR signal from the SbzM reaction with ¹³C, ¹⁵N-labeled cysteine, due to the low sensitivity of ¹⁵N NMR. We also tried to increase the substrate concentration, but the amount of the product did not increase as the enzyme precipitated. As pointed out by the reviewer, judged from the m/z, the ¹³C NMR chemical shift, and the observation that **7** was converted into **5** by aldehyde dehydrogenase SbzJ, we believe that it is reasonable to deduce **7** as a sulfonamide aldehyde.

Line 29 knowledges should be singular

Line 37 analyses should be analysis

Line 119 ezyme

Line 120 sreptomycies

Thank you for the notice. We corrected them.

Line 211 the authors only are talking about Cu in this sentence, and therefore I do not understand what is meant by “each metal”

We deleted “each metal” from the sentence.

We also edited some words, figures, and manuscript and figure titles as the editor suggested.

We found one mistake in the introduction, and revised it as below.
(Introduction line8) “sulfadixiamycin B and C” → “sulfadixiamycin A”

We hope you will agree that the manuscript has been significantly improved, and that you will find it acceptable for publication.

Yours sincerely,

Ikuro Abe, Ph.D.